# Nutritional Value and Antioxidant, Antimicrobial and Cytotoxic Activity of Wild Macrofungi

**DOI:** 10.3390/microorganisms11051158

**Published:** 2023-04-28

**Authors:** Lina Rocío Dávila Giraldo, Claudia Cristina Pérez Jaramillo, Jonh Jairo Méndez Arteaga, Walter Murillo-Arango

**Affiliations:** 1Grupo de Investigación en Productos Naturales, GIPRONUT, Universidad del Tolima, Ibagué 730006, Colombia; lrdavila@ut.edu.co (L.R.D.G.); jmendez@ut.edu.co (J.J.M.A.); 2Laboratorio Socio-Jurídico en Creación e Innovación—IusLab, Departamento de Ciencias Sociales y Jurídicas, Universidad del Tolima, Ibagué 730006, Colombia

**Keywords:** antimicrobial, antioxidant, cytotoxic, mushrooms, nutritional value, MTT assay, Tolima

## Abstract

Macrofungi are among the most promising sources of biologically active natural products with nutritional qualities and therapeutic values. In this work, the nutritional value of nine species of wild macrofungi from Ibague-Tolima (Colombia) was evaluated. In addition the antioxidant, antimicrobial and cytotoxic activities of an ethanol:water (70:30) extract of wild basidiomata were evaluated. The wild mushrooms’ nutritional potential showed that the genus *Pleurotus* and *Lentinus* have the best protein percentages, with 18.4% and 18.5%. The nine extracts evaluated managed to stabilize the two radicals evaluated; however, lower IC_50_ was found for *Phellinus gilvus* and *Ganoderma australe* extracts. The results showed that *Trametes coccinea*, *Pleurotus floridanus* and *Ganoderma australe* extracts were the most effective as antimicrobials, with high inhibition percentages against *Staphylococcus aureus*, *Escherichia coli*, *Pseudomonas aeruginosa* and *Klebsiella pneumoniae*. Antifungal activity results against *Rhizopus oryzae*, *Penicillium* sp. and *Aspergillus niger* showed that the nine extracts were effective at the concentrations tested. Considering cell viability against isolated leukocytes, seven of the nine extracts showed percentages higher than 50% of cell viability. This research describes the nutritional value of nine wild macrofungi in Colombia and their potential for antimicrobial, cytotoxic and antioxidant activity.

## 1. Introduction

Edible and indigenous mushrooms have been used by humans, not only as a source of food, but also for medicinal purposes, due to their valuable content of nutrients and bioactive compounds. They contain a high proportion of carbohydrates (especially fiber) and protein, low levels of lipids (unsaturated fatty acids and other lipids), as well as being a good source of minerals and vitamins, including thiamine, riboflavin, niacin, biotin and ascorbic acid [1]. Considering proteins, mushrooms can produce large quantities of protein in short periods [2]. Macrofungi are recognized as one of the important food items for their significant roles in human health, controlling and modulating many functions of the human body, such as reducing inflammation, improving gut microbiota, impacting the immune system positively and consequently maintaining a state of good health necessary to reduce the risk of diseases such as diabetes, hypercholesterolemia and cancer [3,4,5].

Biologically active compounds from the macrofungi includes terpenoids, polyphenols and polysaccharides, which possess antifungal, antimicrobial, antioxidant, anti-inflammatory, anticancer, antitumor and antiviral properties [6,7,8,9]. Regarding antimicrobial activity, research into natural products has demonstrated significant progress in discovering new compounds with antimicrobial activity. In this context, phytochemical analysis of mushroom extracts revealed their potential to treat infectious diseases and control pathogens due to their variety of active secondary and primary metabolites. As examples, the protein extract of the edible mushroom *Auricularia auricular-judae* had an inhibitory effect on *Staphylococcus aureus*, *Bacillus subtilis*, *Escherichia coli*, *Pseudomonas aeruginosa*, *Klebsiella pneumoniae* and *Candida albicans*, while chitosan extracts of the medicinal mushroom *Ganoderma lucidum* inhibited some Gram-positive and Gram-negative bacteria [10,11,12]. Many pharmaceutical substances such as triterpenes, β-glucans, lectins and L-ergothioneine, with potent and unique health-enhancing properties, have been isolated recently from medicinal macrofungi and distributed worldwide [13,14]. Many of them are not strictly pharmaceutical products (real medicines), but rather represent a novel class of dietary supplements (DSs) or nutraceuticals [15]. They are also known as functional or designer foods, phytochemicals, mycochemicals and biochemopreventatives [16]. These components are extractable from the fungal mycelia or basidiomata and represent an important component of the expanding mushroom biotechnology industry [15].

The potential therapeutic and nutritional implications of macrofungi are enormous, but little research has been undertaken worldwide about this topic. An insufficient number of papers is related to the therapeutic and nutritional value of wild macrofungi. In Colombia, 7273 species of fungi are reported, including macromycetes, lichenized fungi, smuts, rusts and yeasts [17]. However, there are only a few reports on the nutritional value of wild basidiomata of *Ramaria* sp., *Macrolepiota colombiana* and *Lactarius indigo* species associated with areas surrounding oak forests in the northeastern Andes, which are collected and consumed by local connoisseurs in the Iguaque, Colombia [18]. Furthermore wild species of *Pycnoporus sanguineus*, *Lentinus crinitus* and *Pleurotus cf tubarius* have been domesticated in solid culture to obtain basidiomata, being *L. crinitus,* the nutritional characterization and nutritional value of wild basidiomas having been reported in several works [19,20]. Although several edible species are reported in Colombia, very few have been nutritionally characterized. This nutritional characterization together with other food analyses such as the amino acid profile, mineral content and fat content, microbiological and toxicity analysis, would allow detailed information on the nutritional profile and support for its registration and inclusion as food in Colombia. On the other hand, some works on the evaluation of extracts with antioxidant, cytotoxic and antimicrobial activity of wild macrofungi are reported. One of them, related to the evaluation of the antibacterial activity of crude extracts obtained from endophytic wild fungus *Xylaria* sp. and *Diaporthe endophytica* showed promising antimicrobial activity against reference strains of *E. coli* (ATCC 25922) and *S. aureus* (ATCC 25923), [21], as well as the report of the antioxidant and anticancer activity of a water-soluble crude polysaccharide extract isolated from the fruiting bodies of the *Ganoderma* aff. *australe* on the radicals DPPH, ABTS and in the osteosarcoma MG-63 human cell line [22]. Therefore, the objective of this research was to determine the nutritional value and the antioxidant, antimicrobial and cytotoxic activity of nine wild macrofungi species from the region of Tolima-Colombia.

## 2. Materials and Methods

### 2.1. Macrofungi Material

Nine species of wild macrofungi were collected in Ibague (Tolima) and identified as: *Trametes coccinea* (Fr.) Hai J. Li and S.H. He, *Auricularia fuscosuccinea* (Mont.) Henn, *Bovista* sp., *Lentinus* sp, *Irpex rosettiformis* C.C. Chen and Sheng H. Wu, in Chen, Chen and Wu, *Pleurotus floridanus* Singer, Pegler, *Ganoderma australe* (Fr.) Pat., *Phellinus gilvus* (Schwein.) Pat and *Trametes elegans (Spreng.)* Fr. Basidiomata were photographed in situ, removed with a knife, deposited in paper bags with the respective collection number and taken to the laboratory [23]. Morphological identification was made from macroscopic and microscopic characteristics (Figure 1). A macroscopic description using mycological guides was then performed [24,25,26,27] Sections of the basidiomata were prepared in microscope slides with 3% KOH, Red Congo or Cotton Blue and Melzer’s reagent (IKI) [28,29]. The specimens were deposited and preserved in the Fungario Universidad del Tolima (FUT). The study has the Collection permission for access to biological resources with non-commercial purposes (Permiso Marco de Recolección, Resolución 2191 de 2018, Universidad del Tolima, Ibagué, Colombia).

### 2.2. Extract Preparation

Basidiomata were washed and dried in an oven (48 h, 40 °C). Then the basidiomata were stored to determine each species’ nutritional value and prepare the extracts. The extraction was carried out by percolation in a water bath (Memmert, Schwabach, Germany) at 40 °C for 48 h; For this, 20 g of dried basidiomatawere used and cut (crushed) into small pieces that were mixed with a mixture of ethanol: water (70:30) in a 1:20 ratio. Subsequently, the extracts were concentrated in a rotary evaporator, eliminating most of the ethanol, and finally, they were lyophilized, for later use in the antimicrobial, antioxidant and cytotoxic analysis. Subsequently, the extracts were characterized by drop-by-drop chemical tests that allowed us to determine the presence of chemical compounds such as polyphenols (Folin Ciocalteu test), carbohydrates (Molish and Benedict test), flavonoids (Shinoda test), terpenes (Lieberman and Salkowski test), saponins (Foam and Rosenthaler test), tannins (ferric chloride and gelatin-salt) and alkaloids (Tanred, Dragendorff, Valser and Mayer test) [16].

### 2.3. Nutritional Value and Mineral Element Composition

The nutritional value (protein, fat, ash content and dietary fiber) was determined according to AOAC procedures [16]. Macro-Kjeldahl method (N × 6.25) was used to determine the protein content. The crude fat was determined using a Soxhlet apparatus, extracting the sample with petroleum ether. Ash content was estimated by incineration at 600 ± 15 °C for 5 h. Dietary fiber was analyzed by a gravimetric method. Mineral constituents comprising potassium (K), magnesium (Mg), iron (Fe), calcium (Ca), copper (Cu), manganese (Mn), zinc (Zn) and phosphorus (P) were determined by an Atomic Absorption Spectrophotometry (SHIMADZU AA-6300) [30].

### 2.4. Antimicrobial Activity

#### 2.4.1. Antibacterial Activity

An antibacterial assay was performed using the strains *Staphylococcus aureus* (ATCC29213), *Escherichia coli* (ATCC25922), *Pseudomonas aeruginosa* (ATCC27853) and *Klebsiella pneumoniae*, in nutrient broth BHI (BHI, Merck, Darmstadt, Germany).

#### 2.4.2. Antifungal Activity

The antifungal assay was tested with the strains *Rhizopus oryzae, Penicillium* sp. and *Aspergillus niger,* using Sabouraud Dextrose Broth (BD, Merck). The microdilution method currently recommended by the Clinical and Laboratory Standards Institute was used. 

The tests were conducted in a microplate reader UV/VIS (Multiskan GO/UV, Thermo Fischer Scientific, Vantaa, Finland), using 96-well trays. A cell suspension of the bacterium was adjusted to 0.5 McFarland for the antibacterial assay, and a cell suspension of the pathogenic fungi was adjusted to 1 × 10^4^ conidia/mL in sterile water for the antifungal assay. The wells were filled with 50 μL of the bacterial or fungi inoculum, 100 μL of liquid medium (BHI, Merck) and 50 μL of Et-AME at different concentrations. A negative control comprised 100 μL of the medium, 50 μL of the inoculum and 50 μL of sterile water. Contamination controls were also used, which included 200 μL of the medium. Finally, the well trays were incubated for 24 h at 37 °C for bacteria and 72 h at 30 °C in dark conditions for pathogenic fungi. The optical density was measured every 12 h, at 460 nm for bacteria and 595 nm for pathogenic fungi. The percentage of growth inhibition was determined, and the half-maxima inhibitory concentration [31] was calculated. Each sample was measured in triplicate. The mean and standard deviation (*n* = 7) were calculated.

### 2.5. Antioxidant Activity (AOX)

#### 2.5.1. ABTS Radical Cation Decolorization Assay

The ABTS assay was based on the method of [32], slightly modified by [33]. Briefly, a radical solution (3.5 mM ABTS and 1.25 mM potassium persulfate) was prepared in sterile water and left to stand in the dark for 24 h. This solution was then diluted with ethanol at 70% to obtain an absorbance of 0.7 ± 0.02 at 734 nm. For the AOX analysis, it was mixed in the ratio 1:49; extract: radical. It was found to be between 25 and 200 mg/L. The change in optical density was measured in a spectrophotometer (Thermo, Evolution 260) at 734 nm after 6 min. The ABTS scavenging capacity of Et-AME was compared with the standard Trolox curve between 0.0312 and 1 µg/mL. AOX was calculated as the percentage inhibition of absorbance by the following formula: $$AOX_{\mathrm{ABTS}}=\left(\frac{A_{ABTS}-A_{6min}}{A_{ABTS}}\right)\times 100$$
where *A_ABTS_* is the absorbance of ABTS radical in sterile water and *A*_6__*min*_ is the absorbance of ABTS radical solution mixed with sample extract/standard. Each sample was measured in triplicate. The mean and standard deviation were calculated.

#### 2.5.2. DPPH Radical Cation Decolorization Assay

The methodology proposed by [34] was followed with some modifications. A 0.5 mL aliquot of a 0.02% DPPH solution, in ethanol (99%) was added to 0.5 mL of each Et-AME. The mixture was stored in the dark (30 min) and the absorbance at 517 nm against a blank (solvent and DPPH) was measured. The standard curve was prepared with Trolox (0.0039 to 0.0625 μg/mL). The percentage inhibition of DPPH of the test sample and known solutions of Trolox were calculated by the following formula:$$AOX_{\mathrm{DPPH}}=\left(\frac{A_{0}-A}{A_{0}}\right)\times 100$$
where *A*_0_ is the absorbance of DPPH radical in ethanol (blank) and *A* is the absorbance of DPPH radical solution mixed with sample extract/standard. Each sample was measured in triplicate. The mean and standard deviation [2] were calculated.

### 2.6. Cell Viability Measurement Using MTT (3-(4,5-dimethylthiazol-2-yl)-2,5-diphenyltetrazolium Bromide, a Tetrazole) Assay

Blood samples were taken after informed consent was obtained. For this, the leukocytes were separated following the protocol of [35]; four concentrations of the extracts were used (156.25 to 1250 μL/mL). For the assay, microplates were used with 25 μL of leukocytes + 25 μL of Et-AME + 50 μL of MTT were mixed and incubated at 37 °C for 1 h. After this time, DMSO [36] was added, and it was stirred for 10 min to dissolve the formazan crystals. Finally, the microplates were read at 570 nm. This assay was performed in order to assess the viability of the cells [37]. Each sample was measured in triplicate. The mean and standard deviation were calculated.

### 2.7. Statistical Analysis

The data were analyzed using analysis of variance (ANOVA) in order to compare the mean for each concentration and replicates, with a significance level of 95%, using the LSD test. Furthermore, a probabilistic model was conducted to determine the IC_50_ of the Et-AME using Stat graphics^®^ Plus 5.1 statistical program. 

## 3. Results

### 3.1. Qualitative Identification of Secondary Metabolites

Table 1 shows the identification of metabolites in the macromycete extracts. According to the tests mentioned and carried out in Section 2.2, the presence of metabolites such as carbohydrates, polyphenols, flavonoids and terpenes is evident. In the tests carried out, no saponins or alkaloids were detected by the tests that were carried out.

### 3.2. Nutritional Profile

The results of the nutritional and elemental composition of the nine macrofungi are shown in Table 2. The genera *Pleurotus* and *Lentinus* contained an average 18% dry matter protein, followed by *Bovista* sp. with 16%, *I. rosettiformis* with 14.7% and *G. australe* with 12.3%. The fat content was between 7–8%, except for *I. rosettiformis* and *A. fuscosuccinea* with 14.8% and 23%, respectively. A range of 28–53% of total fiber was observed in *G. australe*, *P. gilvus* and *T. elegans*.

The other species showed fiber percentages between 7 and 12%. The mineral’s percentages did not exceed 1%, except for *P. floridanus* and *T. coccinea* which contained 1.7 and 1.24 % of magnesium, respectively. The potassium percentage was between 4.85% and 2%. The ash percentage was in a range of 0.2–4.8%, except for *Bovista* sp. which had 49.71%. 

### 3.3. Antimicrobial Activity

#### 3.3.1. Antibacterial Activity

Only extracts of *T. coccinea*, *G. australe and P. floridanus* showed inhibitory effects for the four bacteria tested. The analysis of variance indicated significant differences between the concentrations evaluated for each of the bacteria tested. Table 3 shows an adjusted model with an average percentage inhibition and prediction of the concentration in mg/L. The extracts that were most effective against *P. aeruginosa*, *S. aureus*, *E. coli* and *S. tiphy*, were *P. floridanus*, *G. australe* and *T. coccinea,* respectively, for each bacterium.

#### 3.3.2. Antifungal Activity

Table 4 shows the adjusted model for the Et-AME tested against three fungal strains (*R. oryzae*, *Penicillium* sp. and *A. niger*) and a dose inhibitory effect. In some cases, the inhibition at the concentrations evaluated was between 80 and 90%, and the data were non-linear. The extract of *I. rosettiformis* showed a 92.98% inhibition against *R. oryzae*. The extract of *T. coccinea* showed the best IC_50_ (22, 30 mg/L) against *R. oryzae*, followed by *Lentinus* sp. with 33.11 mg/L and *T. elegans* with 46.43 mg/L. Most of the extracts showed an inhibition between 85 and 94% against *Penicillium* sp., and *A. niger*, at the concentrations tested. *T.coccinea*, *P. gilvus* and *Bovista* sp. extracts showed a dose-effect relationship against *Penicillium* sp. with IC_50_ values of 148.70 mg/L, 313.2 mg/L and 440.31 mg/L, respectively.

### 3.4. Antioxidant Activity (AOX)

The most effective fungus in stabilizing the ABTS radical was *P. glivus* with an IC_50_ of 27.6 μL/mL, and the least efficient was *P. floridanus* with an IC_50_ of 367.8 μL/mL. For the stabilization of the DPPH radical the most effective was *I. rosettiformis* with IC_50_ 5.9 μL/mL, and the least efficient was a *G. australe* with 4838 μL/mL. Finally, to compare results against a radical stabilizer, Trolox was used obtaining an IC_50_ of 3.15 × 10^−2^ μL/mL (Table 5).

### 3.5. Cell Viability Evaluation

Figure 2 shows the percentages of cell viability of each of the nine fungal extracts analyzed, *P. floridanus* showed the highest cell viability three times higher than the activity shown by *A. fuscosinea* (29%), which was the species that showed the lowest cellular viabilities. The difference between the cell viability of the nine fungal extracts were remarkable, the species *A. fuscoccinea* and *T. coccinea* recording a cell viability lower than 50%; All these results were compared with a positive control which allowed cell viability higher than 80%

## 4. Discussion

### 4.1. Extracts and Chemical Test

As shown in Table 1, metabolites of carbohydrates, phenols, flavonoids, terpenes and alkaloids were detected, highlighting the presence of phenolic and flavonoid metabolites in the fungal extracts, which can be related to AOX in macrofungi. as shown in previous reports [38,39,40].

The sugars were present in all the evaluated extracts, differentiating the species *P. gilvus, G. australe* and *H. palmatus* for having reduced sugars within their metabolites. Polysaccharides are the most abundant in macrofungi and have been widely reported in the scientific literature for their antitumor action [41]. Other studies corroborate that many polysaccharides, specially β-glucans, proteins and polysaccharide-protein complexes define bioactivity in macrofungi [42]. Among the genus Ganoderma, *G. lucidum* has been reported for representative antioxidant action related to the content of polysaccharides [43].

Concerning the presence of terpenes and alkaloids, low concentrations were detected, the presence of alkaloids in most of the species being doubtful, given their non-detection with the majority of colorimetric tests; there is a greater possibility of finding them in *P. gilvus* and *I. rosettiformis*. Saponins were not detected under the conditions performed in the trials.

One of the most productive genera in terms of the amount of triterpenes is the Ganoderma genus, widely used in traditional oriental medicine (Chinese, Japanese, Korean, Vietnamese, etc.) [13]; however, under the tests carried out in this study, its presence was not so marked. This could be due to the type of extraction that we carried out, since we worked with ethanol:water extracts and, according to authors such as [44], this type of compound must be extracted with chloroform and fractionated under pH conditions.

### 4.2. Nutritional Profile

Consumption of macrofungi in Colombia is not widespread. There are reports of some indigenous communities using macrofungi, such as the Huitoto, Muinane and Andoke in the Caqueta and the Inganos community in the Putumayo area [45,46]. In contrast, many rural populations especially in Japan, cultivate various mushroom species. They are used in traditional Japanese cooking and are also well known outside of Japan [15]. 

There are more than 200 genera of fungi, containing species of popular use and various publications report the high-protein content of macrofungi such as *Pleurotus,* with reports of 20–30% of dry biomass. This genus has unique characteristics for its fiber, carbohydrates, vitamins, minerals and high protein content [47]. These protein values could be compared with products such as beans, corn, rice and wheat protein that contain 28%, 10.2%, 7.6% and14.3% protein, respectively [48,49]. These values are close to those obtained in this study for the *Pleurotus* and *Lentinus* species. They are of particular interest because it is the genus most cultivated in different parts of the country and the second genus found after *Agaricus* or Champignons, in many local markets [46]. Other fungi from this study are consumed around the world, including *T. elegans* in the United Republic of Tanzania [50], *A. fuscossinea* in northern Guatemala [51] and also, *I. rosettiformis* in Brazil and México [52].

Another quality of these macrofungi is the low fat content; compared with other foods that have fat content between 1–15% [53]. This research shows that the fat content in the macrofungi analyzed is approximately 7%. This indicates that these macrofungi could be a good food alternative for people with high cholesterol and diabetes [14,24,54]. In addition, edible macrofungi contain several components such as non-digestible fiber, that improve the digestion process [25]. The dietary fiber content and composition in edible mushroom vary greatly with morphological stages including fruit body, mycelium and sclerotium [26] and between different species. The dietary fiber is constituted mainly of water-insoluble fibers, such as chitin and β-glucans [26]. Currently, β-glucans of macrofungi s are used as a food ingredient in baked goods and baking mixes, beverages and beverage bases, cereal and cereal products, dairy product analogs, milk and milk products, plant protein products, processed fruits and fruit juices, soft candy and soup and soup mixes [27,55], because of the therapeutic benefits they have.

The macrofungi tested had varying contents of minerals such as phosphorus, potassium, iron, copper, magnesium and zinc, which depended on the mushroom. The amounts of minerals measured in the fungi from this study are lower than the values reported in other edible fungi [56,57]. These low values could be because the intrinsic characteristic of the natural substrate, for example, dead wood, that has different characteristics depending on the source, and contributed to the mineral content of the macrofungi. Hence, the mineral concentration of the fungi could be useful to predict the type of substrate from which they come [58]. These values could be used in bioremediation of soils or to adjust a liquid or solid medium for optimal growth.

### 4.3. Antimicrobial Activity

The antimicrobial analysis showed that mushroom extracts were bacteriostatic and antifungal and could be postulated as a new natural alternative in the treatment of infections caused by the four bacteria analyzed. Many of these bacteria have become resistant to antibiotics and they are a worldwide problem. They are often isolated from patients with nosocomial infections, and they can also be associated with food diseases [59,60]. Most of the extracts analyzed showed antifungal activity, which could be used as a natural alternative for the control of postharvest decay generated by agents that cause rot (*R. oryzae, Penicillium* sp. *and A. niger*) of various crops including corn, potato, tomato and strawberries.

### 4.4. Antioxidant Activity (AOX)

For the present study, two different assays were performed to evaluate the AOX properties of the Et-AME of the nine fungal species; these were the uptake of radicals by ABTS (stabilization of free radicals through the transfer of hydrogen atoms) and DPPH (free radical stabilization by electron transfer). The extracts revealed a great abundance of secondary metabolites, as can be seen in Table 1, among which we can highlight phenolic compounds, flavonoids and terpenes; which present a wide range of biological effects that include antibacterial, anti-inflammatory, anti-hyperglycemic and antioxidant actions [60]. In addition, different studies have reported phenolic compounds as endogenous antioxidant molecules in the cell [61]. Since they can act as inhibitors of free radicals, peroxide de-composers, metal inactivators, or oxygen scavengers [62]. Aside from phenolic compounds, relatively high levels of vitamins A, C and carotene in macrofungi have been shown to be key contributors to their antioxidant activity [63].

In studies such as those carried out by [60,61,62,63], the antioxidant potential of some species of macromycetes, including *Agaricus bisporus*, *Boletus badius* and *Hericium erinaceus,* is highlighted, as well as *Hypsizigus marmoreus*, *Lentinula edodes*, *Lepista nuda*, *Pleurotus* sp., *Polyporus squamosus*, *Russula delica*, *Termitomyces* sp., *Volvariella volvacea* and *Verpa conica*, highlighting said potential by the presence of metabolites such as polysaccharides and polyphenols [64].

The antioxidant activity shown by the nine native species evaluated in this work by the two synthetic methods DDPH and ABTS was different, and the two methods showed us individuals with greater stabilization capacity as can be seen in Table 5, which demonstrates the diversity of activities that can be generated through currently unknown species of macrofungi.

### 4.5. Cell Viability Evaluation

Although blood cells tend to be very prone to damage, they can come to express important preliminary results to know the toxicity that the extracts under study may present. In Figure 2, the concentrations necessary for the survival of 50% of the cells evaluated were determined. Significant differences were found between the nine species of macrofungi, for example, in the IC_50_ obtained from the extracts of *A. fuscoccinea*, *G. australe* and *T. coccinea* [64]. The low toxicity shown by most of the evaluated species may be related to the type of metabolites determined in this study, where carbohydrates, phenols and flavonoids stand out. In addition, it could be assumed that the metabolites found are found in low concentrations, which decreases the toxicity shown and makes them suitable for consumption [65]. In contrast, in macrofungi such as *Bovista* sp., *T. elegans* and *P. floridanus*, a high IC_50_ was obtained, which could be due to the presence of terpene-type compounds, which contribute to biomembrane damage [66].

Terpene compounds and derivatives are widely distributed in angiosperms, bryophytes and fungi, and perform defense functions in all plant and fungal individuals. However, they also confer high toxicity, thus generating antitumor, anti-inflammatory, antimalarial and antimicrobial activities, among others [66]. Some species of the genus Bovista show the presence of several sesquiterpenes with high rates of cytotoxicity [67]. There are many species with cytotoxic characteristics. One of them, Lentinus edodes, is an edible with has special characteristics; some people may experience minor side effects or allergic reactions. The most serious allergic reactions are those to its spores that have been reported in collectors; symptoms include fever, headache, congestion, cough, sneezing, nausea and general malaise. Whit an extract of the fruiting body the decreased effectiveness of blood platelets in initiating coagulation was found [68] The cytotoxicity shown by *Bovista* sp., *T. elegans* and *P. floridanus* in this study shows us their possible potential for the treatment of degenerative diseases or even cancer. Results that can be related to studies such as the one carried out by [65] in which the toxicity of polysaccharides isolated from *Trametes versicolor* against cancer cells was demonstrated.

## 5. Conclusions

This research analyzed nine native mushroom species for their nutritional value, antimicrobial and antioxidant activity, and cell cytotoxicity. The results show that these native species had valuable nutritional potential, in particular *P. floridanus* and *Lentinus* spp., due to the contribution of proteins, lipids and minerals of their basiodiomata. In addition, *G. australe*, due to its high fiber content (over 50%), turns out to be of excellent nutraceutical value for use in functional foods or additives in foods. On the other hand, the Et-AME inhibited the growth of pathogenic fungi showing an inhibition between 85% and 94% against *Penicillium* sp., and *A. niger*, whilst at the concentrations tested, only extracts of *T. coccinea, G. australe* and *P. floridanus* showed inhibitory effects for the four bacteria tested. This opened the possibility of finding new natural alternatives to treating infections with these mushrooms. The most effective fungus in stabilizing the ABTS radical was *P. glivus*, and the least efficient was *P. floridanus*, and *I. rosettiformis* had a good stabilization of the DPPH radical. In addition, as well as the difference between the cell viabilities of the nine fungal extracts, which were remarkable, *A. fuscoccinea* and *T. coccinea* recorded a cell viability lower than 50%, indicating that they can be analyzed in more detail on cancer cell lines. The results of this study show the potential use of the nutritionally and bioactive species studied; however, it is necessary to perform domestication assays and use other biotechnological strategies to grow these specimens under controlled conditions.

## Figures and Tables

**Figure 1 microorganisms-11-01158-f001:**
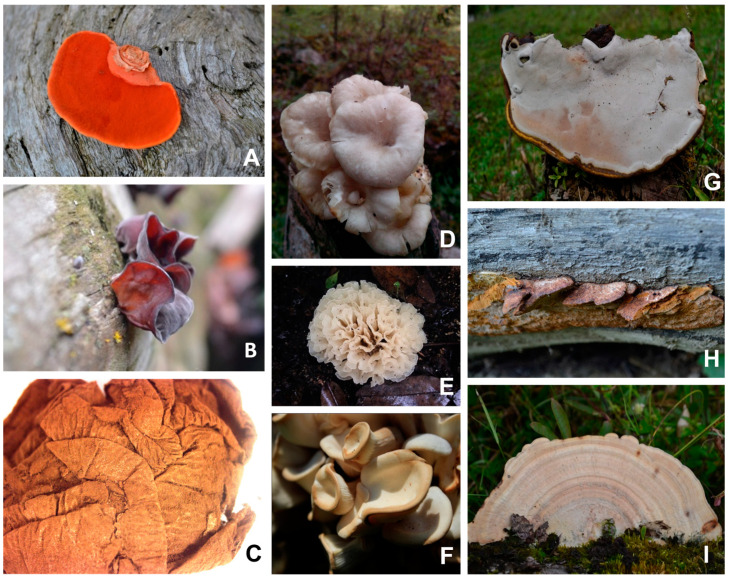
Wild macrofungi collected for this study. (**A**) Basidiomata of *Trametes coccinea*. (**B**) Basidiomata of *Auricularia fuscosuccinea*. (**C**) Basidiomata of *Bovista* sp. (**D**) Basidiomata of *Lentinus* sp. (**E**) Basidiomata of *Irpex rosettiformis*. (**F***)* Basidiomata of *Pleurotus tubarius*. (**G**) Basidiomata of *Ganoderma australe*. (**H**) Basidiomata of *Phellinus gilvus*. (**I**) Basidiomata of *Trametes elegans*.

**Figure 2 microorganisms-11-01158-f002:**
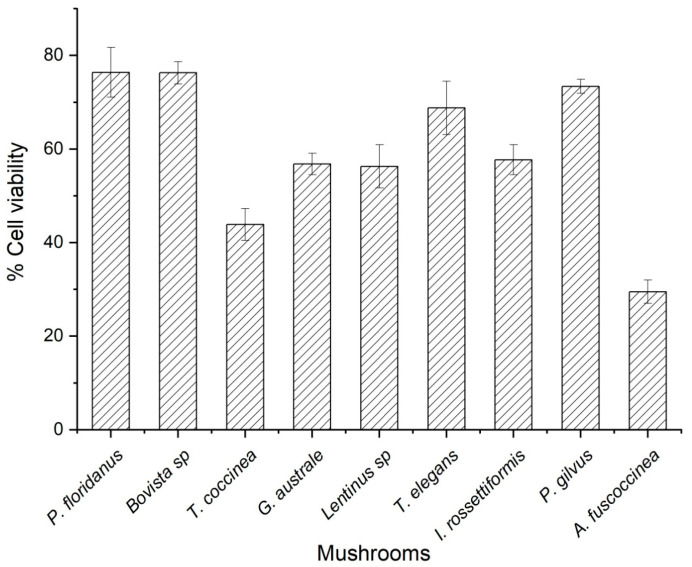
Cell viability as percentage of the Et-AME.

**Table 1 microorganisms-11-01158-t001:** Qualitative identification of metabolite groups in Et-AME of the wild macrofungi.

Metabolite	Test	*P. floridanus*	*Bovista* sp.	*T. coccinea*	*G. australe*	*Lentinus* sp.	*T. elegans*	*I. rosettiformis*	*P. gilvus*	*A. fuscoccinea*
Carbohydrate	Benedict	+	-	-	+	-	-	+	+	-
	Molish	+	+	+	+	+	+	+	+	+
Saponins	Foam	-	-	-	-	-	-	-	-	-
	Rosenthaler	-	-	-	-	-	-	-	-	-
Polyphenols	Folin-Ciocalteu	+	-	-	+	+	+	+	+	+
Tannins	Ferric Chloride	-	-	-	-	-	-	-	-	-
	Gelatin-salt	-	-	-	-	-	-	-	-	-
Flavonoids	Shinoda (Zn)	+	+	+	+	+	+	+	+	+
Terpenes	Lieberman	+	+	+	+	+	+	+	+	+
	Salkowski	+	-	+	+	+	+	+	+	+
Alkaloids	Tanred	-	-	+	+	+	+	+	+	+
	Dragendorff	+	+	+	+	+	+	+	+	+
	Valser	-	+	-	-	-	-	+	-	-
	Mayer	-	-	-	-	-	-	-	-	-

(+) Presence; (-) Absence.

**Table 2 microorganisms-11-01158-t002:** Proximate and mineral analysis of spent substrate and wild macrofungi (dry basis %, mg/kg, *w*/*w*).

Macrofungi	% Nitrogen	% Protein	% Total Fiber	% Ether Extract	% Ash	% Humidity	% Ca	%Mg	Na (mg/kg)	% K	Fe (mg/kg)	Cu (mg/kg)	Mn (mg/kg)	Zn (mg/kg)
*P. floridanus*	3.0	18.5	7.8	7.7	4.8	65.2	0.1	1.7	186.2	4.8	78.1	28.7	106.0	45.9
*Bovista* sp.	2.6	16.1	14.3	9.3	49.7	29.4	0.3	0.4	435.9	0.9	568.5	90.5	58.4	93.5
*T. coccinea*	1.6	9.9	10.1	9.3	2.8	7.7	0.2	1.2	1170.0	2.0	72.0	16.6	100.5	12.4
*G. australe*	2.0	12.3	53.6	8.6	1.3	53.5	0.1	0.4	494.6	1.9	75.0	15.3	1.4	23.3
*Lentinus* sp.	2.9	18.4	11.9	9.5	6.0	20.2	0.1	0.5	71.9	0.6	67.2	16.5	1.4	8.2
*T. elegans*	1.0	6.2	28.1	7.6	1.0	34.0	0.1	0.3	83.9	0.3	62.8	14.2	1.4	4.0
*I. rosettiformis*	2.4	14.7	10.7	14.8	3.5	44.1	0.4	0.4	439.5	0.9	18.6	14.7	8.5	15.8
*P. gilvus*	1.2	7.8	31.6	8.5	3.0	8.5	0.1	0.3	212.6	2.6	10.0	15.8	6.0	27.0
*A. fuscoccinea*	1.6	10.0	7.17	22.2	3.0	59.5	0.1	0.8	197.8	0.7	18.5	19.2	6.4	18.3

**Table 3 microorganisms-11-01158-t003:** Percentage of growth inhibition on bacterial strain after 24 h of incubation with the AME.

Bacterium	Concentration (mg/mL)	2500	1250	625	312.5	156.125	78.125	39.05
*E. coli*	*T. coccinea*	438.1 ± 10.9 a	262.1 ± 31.4 b	111.3 ± 47.4 c	76.6 ± 8.1 cd	61.3 ± 39.2 cd	66.3 ± 5.0 cd	58.1 ± 3.6 d
	*P. floridanus*	ND	ND	ND	61.5 ± 4.4 a	55.4 ± 4.2 ab	47.8 ± 7.6 bc	45.2 ± 6.9 c
	*P. gilvus*	ND	ND	ND	ND	51.5 ± 9.3 a	45.5 ± 6.5 a	11.4 ± 3.6 b
	*T. elegans*	43.8 ± 13.8 a	13.0 ± 5.9 b	22.3 ± 8.6 b	22.4 ± 6.8 b	ND	ND	ND
	*Bovista* spp.	ND	ND	ND	ND	ND	ND	ND
	*Lentinus* sp.	ND	ND	ND	61.5 ± 4.4 a	55.4 ± 4.2 ab	47.8 ± 7.6 bc	45.2 ± 6.9 c
	*G. australe*	113.9 ± 36.5 a	35.9 ± 24.4 b	50.6 ± 24.0 b	42.3 ± 9.5 b	40.1 ± 10.4 b	36.8 ± 15.7 b	30.4 ± 5.9 b
	*I. rosettiformis*	ND	ND	ND	ND	ND	ND	ND
	*A. fuscosscinea*	ND	ND	ND	ND	42.8 ± 12.7 a	44.3 ± 16.7 a	43.6 ± 20.7 a
*K. pneumoniae*	*T. coccinea*	164.4 ± 10.3 a	102.7 ± 3.3 b	72.9 ± 4.5 c	52.8 ± 1.6 d	ND	ND	ND
	*P. floridanus*	27.1 ± 1.1 b	33.7 ± 1.6 a	33.9 ± 1.6 a	27.0 ± 3.5 b	ND	ND	ND
	*P. gilvus*	49.1 ± 1.3 a	41.8 ± 1.1 b	44.3 ± 0.7 b	38.3 ± 3.3 c	ND	ND	ND
	*T. elegans*	38.4 ± 5.8 a	32.6 ± 2.3 ab	19.8 ± 7.9 bc	24.4 ± 9.2 c	ND	ND	ND
	*Bovista* spp.	45.8 ± 13.5 a	32.6 ± 2.7 ab	44.4 ± 6.3 bc	27.9 ± 5.4 c	ND	ND	ND
	*Lentinus* sp.	35.7 ± 1.7 a	36.4 ± 1.4 a	21.3 ± 3.0 b	28.7 ± 2.9 b	ND	ND	ND
	*G. australe*	51.4 ± 2.2 a	34.1 ± 6.5 b	35.7 ± 1.3 b	35.2 ± 1.8 b	ND	ND	ND
	*I. rosettiformis*	33.5 ± 1.6 a	30.0 ± 0.8 b	27.9 ± 1.4 b	24.5 ± 2.3 c	ND	ND	ND
	*A. fuscosscinea*	38.7 ± 2.9 a	37.9 ± 2.0 a	30.9 ± 1.4 b	21.1 ± 1.2 c	ND	ND	ND
*P. aeruginosa*	*T. coccinea*	128.3 ± 3.5 a	73.8 ± 4.3 b	58.7 ± 9.5 c	43.3 ± 2.1 d	ND	ND	ND
	*P. floridanus*	48.2 ± 5.2 a	39.8 ± 15.3 a	24.9 ± 3.0 b	18.0 ± 3.8 b	ND	ND	ND
	*P. gilvus*	35.7 ± 5.3 a	29.2 ± 6.9 ab	31.9 ± 2.5 a	24.5 ± 0.8 b	ND	ND	ND
	*T. elegans*	10.8 ± 8.2 a	6.9 ± 3.3 a	4.3 ± 3.7 a	10.4 ± 4.7 a	ND	ND	ND
	*Bovista* spp.	8.8 ± 5.0 a	12.1 ± 2.8 a	16.1 ± 3.3 a	13.6 ± 6.2 a	ND	ND	ND
	*Lentinus* sp.	14.0 ± 0.9 a	9.1 ± 1.4 b	7.1 ± 0.9 c	2.5 ± 1.3 d	ND	ND	ND
	*G. australe*	17.5 ± 3.1 ab	20.9 ± 2.1 a	19.4 ± 1.8 ab	16.3 ± 1.6 b	ND	ND	ND
	*I. rosettiformis*	16.7 ± 1.7 a	14.9 ± 1.4 a	12.1 ± 0.8 b	10.7 ± 1.2 b	ND	ND	ND
	*A. fuscosscinea*	20.5 ± 6.1 a	17.0 ± 6.5 ab	10.7 ± 1.8 bc	6.1 ± 3.1 c	ND	ND	ND
*S. aureus*	*T. coccinea*	407.8 ± 17.0 a	238 ± 14.9 b	164.8 ± 11.0 c	91.7 ± 15.5 d	39.0 ± 4.2 e	36.1 ± 11.2 e	ND
	*P. floridanus*	48.1 ± 17.7 a	43.3 ± 18.9 a	27.4 ± 14.8 ab	9.8 ± 3.5 b	ND	ND	ND
	*P. gilvus*	55.8 ± 8.2 a	71.5 ± 21.1 a	58.3 ± 10.7 a	16.8 ± 4.1 b	ND	ND	ND
	*T. elegans*	65.4 ± 15.6 a	43.2 ± 5.0 b	ND	ND	ND	ND	ND
	*Bovista* spp.	ND	ND	ND	ND	ND	ND	ND
	*Lentinus* sp.	23.4 ± 15.1 a	31.3 ± 15.2 a	ND	ND	ND	ND	ND
	*G. australe*	45.7 ± 8.6 a	53.1 ± 22.1 a	52.7 ± 5.3 a	51.0 ± 9.3 a	48.3 ± 6.7 a	ND	ND
	*I. rosettiformis*	ND	ND	ND	ND	ND	ND	ND
	*A. fuscosscinea*	27.5 ± 13.9 a	18.7 ± 11.3 a	14.7 ± 8.7 a	ND	ND	ND	ND
Oxytetracycline hydrochloride	Concentration (mg/mL)	400	200	100	50			
*E. coli*		99.04 ± 0.48 a	99.63 ± 0.46 a	97.17 ± 0.49 b	94–66 ± 0.33 c			
*K. pneumoniae*		79.17 ± 2.44 a	63.14 ± 3.31 b	58.44 ± 1.35 c	52.25 ± 1.94 d			
*P. aeruginosa*		99.19 ± 0.25 a	99.01± 0.18 a	98.42 ± 0.06 b	97.37± 0.18 c			
*S. aureus*		96.13 ± 0.95 a	92.04 ± 1.52 b	90.42 ± 1.52 b	89.53 ± 3.00 b			

Data are expressed as mean ± sd (*n* = 7). (Bonferroni method). ND: —not determined.

**Table 4 microorganisms-11-01158-t004:** Percentage of growth inhibition (INH) of pathogenic fungi after 48 h of incubation with the Et-AME.

Fungi	Concentration (mg/mL)	312.5	156.25	78.125	39.063
*Aspergillus niger*	*T. coccinea*	91.2 ± 6.5 a	86.6 ± 30 ab	86.3 ± 2.4 ab	81.9 ± 3.5 b
	*P. floridanus*	83.7 ± 6.7 a	82.4 ± 5.9 a	77.4 ± 9.9 a	77.1 ± 5.1 a
	*P. gilvus*	85.8 ± 2.4 a	80.0 ± 6.4 a	77.8 ± 7.1 a	43.4 ± 19.9 b
	*T. elegans*	86.7 ± 1.8 a	87.4 ± 1.6 a	85.0 ± 3.7 a	69.6 ± 13.2 b
	*Bovista* spp.	87.0 ± 2.4 a	84.6 ± 3.1 a	78.3 ± 7.2 a	65.1 ± 10.3 b
	*Lentinus* sp.	88.0 ± 2.1 a	84.9 ± 8.3 a	91.1 ± 2.7 a	88.1 ± 4.6 a
	*G. australe*	83.0 ± 2.8 a	83.0 ± 2.9 a	81.5 ± 2.6 a	80.0 ± 1.4 a
	*I. rosettiformis*	92.6 ± 4.2 a	95.7 ± 3.5 a	96.0 ± 2.3 a	51.7 ± 11.4 b
	*A. fuscosscinea*	87.9 ± 3.5 a	87.3 ± 3.9 a	77.5 ± 15.2 a	77.9 ± 1.8 a
*Penicillium* sp.	*T. coccinea*	93.7 ± 2.1 a	84.2 ± 5.4 b	90.1 ± 4.4 ab	92.01 ± 3.9 a
	*P. floridanus*	92.0 ± 5.3 a	91.2 ± 11.1 a	90.0 ± 5.6 a	94.6 ± 5.9 a
	*P. gilvus*	93.4 ± 3.9 a	96.6 ± 4.1 a	91.7 ± 4.5 a	59.3 ± 12.2 b
	*T. elegans*	90.8 ± 8.8 a	90.2 ± 4.8 a	82.3 ± 1.3 a	85.3 ± 3.8 a
	*Bovista* spp.	91.3 ± 5.1 a	88.4 ± 2.7 a	90.9 ± 10.7 a	78.4 ± 17.6 a
	*Lentinus* sp.	96.5 ± 6.0 a	86.0 ± 3.6 b	87.9 ± 2.5 b	93.1 ± 5.1 ab
	*G. australe*	91.2 ± 1.0 a	87.5 ± 4.5 a	86.7 ± 4.3 a	88.7 ± 3.4 a
	*I. rosettiformis*	90.9 ± 5.7 a	88.2 ± 3.7 a	86.7 ± 3.9 a	67.3 ± 17.2 b
	*A. fuscosscinea*	89.6 ± 4.8 a	91.3 ± 3.2 a	95.2 ± 4.7 a	93.0 ± 6.0 a
*Rhizophus oryzae*	*T. coccinea*	86.0 ± 7.8 a	82.0 ± 9.8 a	79.9 ± 6.7 a	74.0 ± 10.1 b
	*P. floridanus*	71.8 ± 6.5 a	50.7 ± 15.8 b	32.3 ± 4.6 c	27.2 ± 7.4 c
	*P. gilvus*	27.3 ± 3.1 a	19.6 ± 1.5 b	14.7 ± 1.5 b	14.3 ± 2.4 b
	*T. elegans*	48.4 ± 7.4 a	48.6 ± 7.5 ab	34.0 ± 8.7 b	29.4 ± 10.2 b
	*Bovista* spp.	31.8 ± 6.3 a	32.1 ± 5.3 b	27.8 ± 6.0 b	25.9 ± 5.3 b
	*Lentinus* sp.	87.2 ± 6.9 a	86.7 ± 11.4 a	63.3 ± 13.4 b	54.8 ± 8.5 b
	*G. australe*	46.7 ± 9.6 a	52.8 ± 7.1 ab	40.9 ± 11.7 b	39.3 ± 7.8 b
	*I. rosettiformis*	91.3 ± 3.1 a	94.9 ± 2.1 a	92.8 ± 3.9 a	20.9 ± 6.6 b
	*A. fuscosscinea*	28.05 ± 2.1 a	30.7 ± 4.5 b	26.5 ± 6.2 b	28.8 ± 4.6 b

Data are expressed as mean ± sd (*n* = 7). (Bonferroni method).

**Table 5 microorganisms-11-01158-t005:** IC_50_ values for DPPH and ABTS radicals stabilization.

	ABTS	DPPH
Macrofungi	IC50 (mg/L)	IC50 (mg/L)
*T. coccinea*	77.9 ± 2.4 a	1522.4 ± 1.4 c
*P. floridanus*	119.5 ± 1.2 a	450.1 ± 0.7 b
*P. gilvus*	36.8 ± 1.8 a	293.5 ± 5.6 b
*T. elegans*	175.7 ± 5.3 b	1402.7 ± 6.6 c
*Bovista* spp.	256.8 ± 4.7 b	5471.2 ± 7.7 d
*Lentinus* sp.	57.4 ± 1.1 a	4402.3 ± 5.5 d
*G. australe*	110.3 ± 4.9 a	88.3 ± 5.6 ab
*I. rosettiformis*	224.7 ± 7.8 b	4412.5 ± 8.8 d

Data are expressed as mean ± sd (*n* = 7). (Bonferroni method, letters a–d, significant differences between means).

## Data Availability

Not applicable.

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
