# Peer review of "Nutritional Value and Antioxidant, Antimicrobial and Cytotoxic Activity of Wild Macrofungi"

_microorganisms, 2023, doi:10.3390/microorganisms11051158_

Round 1
Reviewer 1 Report
This manuscript deals with the bioactivities of the ethanol extracts of 9 mushroom species from Columbia. The author investigated the chemical composition, antifungal, antibacterial, antioxidant and cytotoxic activities of the extracts in a systematic manner. However, there is a lack of information in the methodology used. Data interpretation also seems confusing, affecting the overall impact of the findings presented herein.
1. Methodology used must be clear and contains sufficient details to be replicated.
Lines 55-61: who authenticate the mushroom species and by what method?
Lines 112: what is "(0,0312 a 1 µg/mL)"?
Line 136: ethics approval should be included here
Line 140: the duration of the treatment was only 1 hour?
2. In Table 2, SD is missing.
3. Positive controls for antibacterial, antifungal and antioxidant activities are missing. Line 211: what was the positive control used in the experiment?
4. Table 5: the IC50 values of the extracts in the range of mgml are considered too high, by comparison to literature data, these extracts deemed to be devoid of antioxidant activity.
5. Lines 232-237: polysaccharides are unlikely to be present in aqueous ethanol extracts, hence, the citations on polysaccharides seem irrelevant to this section
6. The discussion section is too long and lacked focus. The authors should compare and contrast their results with those in the literature. Are their results consistent with previous work done on the same species?
7. The conclusion should be rephrased to reflect the overall findings of this study rather than on the postulations.
Author Response
Dear reviewer, the manuscript has been modified according to your request. Below are the changes made to the manuscript
- Methodology used must be clear and contains sufficient details to be replicated.
-Lines 55-61: who authenticate the mushroom species and by what method?
The procedure is described in the methodology in lines 89 to 99. The information about this topic was completed. In addition, one of the authors is qualified in the mushroom taxonomy. figure 1, containing photographs of specimens, was added too
-Lines 112: what is "(0,0312 a 1 µg/mL)"? This value refers to the concentrations of the standard curve of Trolox
Line 136: ethics approval should be included here. Ethics approval was included in lines 99 to 101.
Line 140: the duration of the treatment was only 1 hour? Yes, for this kind of cells, the assay takes one hour. The reference protocol is described in “Oez, S., Platzer, E. & Welte, K. A quantitative colorimetric method to evaluate the functional state of human polymorphonuclear leukocytes. Blut 60, 97–102 (1990). https://doi.org/10.1007/BF01720515”
- In Table 2, SD is missing. Table 2 corresponds to the proximate and mineral analysis of spent substrate and wild mushrooms. The analyses were performed one time only.
- Positive controls for antibacterial, antifungal and antioxidant activities are missing. Line 211: what was the positive control used in the experiment? We added Positive control for antibacterial, antifungal, and antioxidants activities.
- Table 5: the IC50 values of the extracts in the range of mgml are considered too high, by comparison to literature data, these extracts deemed to be devoid of antioxidant activity. The values we reported as IC50 in the different trials are expressed in mg/L, µg/mL, but not in mg/ml.
- Lines 232-237: polysaccharides are unlikely to be present in aqueous ethanol extracts, hence, the citations on polysaccharides seem irrelevant to this section. Based on our experience, we can affirm that polysaccharides can also be detected in both aqueous extracts and ethanol-water. Even in the literature, polysaccharides are reported and extracted in water, but also ethanol-water mixtures, although they are not as frequent.
- The discussion section is too long and lacked focus. The authors should compare and contrast their results with those in the literature. Are their results consistent with previous work done on the same species? We consider that the discussion does not extend too long since it was necessary to expand a little due to the volume of results. On the other hand, unlike the genera Ganoderma and Lentinus, the other species studied do not have detailed information on their nutritional and bioactive potential.
- The conclusion should be rephrased to reflect the overall findings of this study rather than on the postulations. The conclusion was changed according to your request
Reviewer 2 Report
The title should be changed.
The introduction is too short, also add some more references.
In 2.2 the dates of sampling, year, date and place are missing.
The structure of the Results section is not appropriate, there are several tables without text, this should be improved.
References should be added.
Author Response
Dear reviewer, the manuscript has been modified according to your request
The title should be changed. The title was modified
The introduction is too short, also add some more references. The introduction was completed according to your request
In 2.2 the dates of sampling, year, date and place are missing. The dates were added
The structure of the Results section is not appropriate, there are several tables without text, this should be improved. The results, especially the tables were improved
References should be added. New references was added to the manuscript
Round 2
Reviewer 2 Report
Accept the work in the present form, please chech the references again.